# Aquatic Biomaterial Repositories: Comprehensive Guidelines, Recommendations, and Best Practices for Their Development, Establishment, and Sustainable Operation

**DOI:** 10.3390/md22090427

**Published:** 2024-09-20

**Authors:** Christiana Tourapi, Eleni Christoforou, Susana P. Gaudêncio, Marlen I. Vasquez

**Affiliations:** 1Department of Chemical Engineering, Cyprus University of Technology, Archiepiskopou Kyprianou 30, 3036 Limassol, Cyprus; ctourapi@yahoo.com (C.T.); eleni.christoforou@cut.ac.cy (E.C.); 2Associate Laboratory i4HB, Institute for Health and Bioeconomy, NOVA Faculty of Sciences and Technology, NOVA University of Lisbon, 2819-516 Lisbon, Portugal; s.gaudencio@fct.unl.pt; 3Research Unit on Applied Molecular Biosciences, UCIBIO, Chemistry Department, NOVA Faculty of Sciences and Technology, NOVA University of Lisbon, 2819-516 Lisbon, Portugal

**Keywords:** blue biotechnology, bioprospection, marine bioresources, marine natural products, biorepositories, biobanks

## Abstract

The alarming pace of species extinction severely threatens terrestrial and aquatic ecosystems, undermining the crucial ecological services vital for environmental sustainability and human well-being. Anthropogenic activities, such as urbanization, agriculture, industrialization, and those inducing climate change, intensify these risks, further imperiling biodiversity. Of particular importance are aquatic organisms, pivotal in biodiscovery and biotechnology. They contribute significantly to natural product chemistry, drug development, and various biotechnological applications. To safeguard these invaluable resources, establishing and maintaining aquatic biomaterial repositories (ABRs) is imperative. This review explores the complex landscape of ABRs, emphasizing the need for standardized procedures from collection to distribution. It identifies key legislative and regulatory frameworks, such as the Nagoya Protocol and EU directives, essential for ensuring responsible and equitable biorepository operations. Drawing on extensive literature and database searches, this study compiles existing recommendations and practices into a cohesive framework with which to guide the establishment and sustainable management of ABRs. Through collaborative efforts and adherence to best practices, ABRs can play a transformative role in the future of marine biotechnology and environmental conservation.

## 1. Introduction

Over 38,500 species, both aquatic and terrestrial, are currently threatened with extinction. This represents an astounding 28% of the 138,374 species that have been assessed by The International Union for Conservation of Nature (IUCN) Red List of Threatened Species [1]. The invaluable services organisms provide to environmental health may well disappear along with these species. The direct and indirect causes of the loss of biodiversity and ecosystems are (i) residential and commercial development; (ii) agriculture and aquaculture; (iii) energy production and mining; (iv) transportation and service corridors; (v) biological resource use; (vi) human intrusions and disturbance; (vii) natural system modifications; (viii) invasive and other problematic species, as well as genes and diseases related to these; (ix) pollution; (x) geological events; and (xi) climate change and severe weather conditions [2]. In particular, aquatic organisms play a major role in environmental and human health and are highly significant for biodiscovery and biotechnology. The significance of aquatic biota to human society is demonstrated by their contribution to the research fields of natural product chemistry and drug discovery. Bioresources have been shown to produce highly diverse biomolecules, unique structures, and bioactivities that open new avenues for the bioeconomy, inspiring the production of natural drugs (NDs). They have other biotechnological applications, acting as sources of food, nutraceuticals, fertilizers, cosmetics, cosmeceuticals, and textiles [3,4,5,6]. Thus, protecting aquatic ecosystems and establishing biorepositories for biomaterials and bioactive compounds is crucial.

The establishment and sustainable operation of an ABR is a multilayered process that requires knowledge, expertise, specialized personnel, and novel technological, biochemical, and molecular applications. Within this study, an aquatic biomaterial repository (ABR) refers to an organization that collects, processes, stores, and distributes materials derived from aquatic biospecimens, i.e., extracted through the application of specialized genomic and chemical methods to DNA and RNA, proteins, enzymes, biopolymers, and secondary metabolites (biomolecules and natural products). An ABR may include physical samples or biospecimens obtained from various sources: (i) from natural history collections (NHCs); (ii) from bioprospecting and biodiscovery sampling; (iii) from relevant research activities by academic or research institutions; (iv) from the cultivation of biospecimens; and (v) from university-based or private collections [3].

According to a survey conducted in 2014 [7], noticeable variations in practices and methods used for collection, sampling, processing, storage, and distribution were detected among different institutions, and among biodiversity and environmental biobanks (BEBs) [8]. For the past decade, researchers and innovators from numerous life sciences fields have requested the improvement and standardization of research and scientific operating procedures [9]. Without consistency, the nomenclature of BEBs varies [3]; the terms “biobanks” and “biorepositories” are often used interchangeably, and other reported names include environmental repositories, specimen banks, and biospecimen repositories, among others. Moreover, internal policies are not widely disseminated within the scientific community, and each entity drafts its own guidelines. The absence of comprehensive and inclusive published guidelines has generated a gap in terms of creating and standardizing the operation of environmental and biomaterial repositories. In response to this issue, the International Society for Biological and Environmental Repositories (ISBER) published the fourth edition of “ISBER Best Practices: Recommendations for Repositories” in 2018 [10] to ‘harmonize the scientific, technical, legal, and ethical issues relevant to repositories of biological and environmental specimens’ [11].

This study aims to address this issue, to some extent, by providing a concise yet informative document for the establishment and operation of ABRs in Europe and beyond. This document seeks to bridge the gap in standardized guidelines for ABRs by compiling the best practices for each workflow stage: collection, processing, storage, management, and distribution. Serving as a valuable reference point, it assembles available recommendations, guidelines, and best practices related to establishing, operating, and managing an ABR. By providing these insights, the document can incentivize local authorities to establish such guidelines for biorepository development, thereby addressing the loss of aquatic biota and their ecosystems. This effort not only promotes conservation at the local level but also strengthens the local economy and communities.

Given the complexity and interdisciplinary nature of operating an ABR, we conducted comprehensive literature searches, encompassing scholarly and non-scholarly sources. Additionally, we conducted searches into the legislation and legal obligations relevant to ABRs.

Two databases (PubMed and Google Scholar) and one registry (ScienceOpen) were utilized for this review. In PubMed, the search utilized the keywords “marine AND environmental AND specimen bank”, combined with “environment OR environmental AND biospecimen OR biospecimens AND repositories OR repository,” yielding 88 results. Google Scholar searches employed the keywords “marine environmental specimen bank” AND “marine biorepositories.” In the ScienceOpen registry, queries focused on “environmental repositories” AND “marine environmental specimen bank”. The non-scholarly literature search yielded 59 unique results, with 26 obtained from websites. All searches were conducted in April 2023. A more systematic approach was employed to ensure replicability in retrieving documents related to aquatic biomaterial repositories.

## 2. Legislation and Regulatory Tools

To explore recommendations, guidelines, and best practices for aquatic biomaterial repositories, this review examines legislation, regulatory frameworks, and general principles governing their establishment, operation, and management.

It is crucial not to overlook the legislative and regulatory frameworks necessary for the rational establishment and operation of ABRs. Understanding these frameworks and adhering to Responsible Research and Innovation (RRI) principles nationally and internationally are imperative [12]. In cases where aquatic biospecimens and biomaterials are sourced from other countries for research or biotechnological purposes, it is essential for both the biomaterial repository management and the providers to be aware of and comply with relevant legislation. Non-compliance may lead to sanctions, fines, and potentially the cessation of research or commercial activities [3].

The key legislative and regulatory tools that should guide biorepository activities, ensuring equity, ethical conduct, environmental responsibility, and overall accountability, are comprehensively listed in Table 1. All the abbreviations mentioned in this review are listed in Appendix A. Herein summarized as international legislative tools and Other European Union legislative tools, these legislative frameworks collectively aim to protect biodiversity, regulate the use and trading of genetic resources and species, ensure sustainable practices, and promote international cooperation for environmental conservation. International legislative tools include the (i) Nagoya Protocol, which ensures the fair and equitable sharing of benefits from genetic resources [12,13]; (ii) the Cartagena Protocol, which ensures the safe handling, transportation, and use of living modified organisms (LMOs); (iii) the Convention on Biological Diversity (CBD), which aims to promote the conservation of biological diversity, the sustainable use of related resources, and the fair sharing of benefits; (iv) the United Nations Convention on the Law of the Sea (UNCLOS), which establishes regulations on territorial seas, exclusive economic zones, marine environmental protection, and marine scientific research; (v) the Convention on International Trade in Endangered Species (CITES), which regulates international trade in endangered species to prevent over-exploitation; (vi) the Convention on the Conservation of Migratory Species (CMS), which protects migratory species from becoming endangered; and (vii) the Bern Convention, which conserves European wildlife and natural habitats, focusing on endangered and vulnerable species. Other European Union legislative tools include the (i) EU Marine Strategy Framework Directive (MSFD), which protects marine ecosystems and biodiversity, helping EU countries to achieve a good environmental status (GES); (ii) the EU Water Framework Directive (WFD), which protects aquatic ecology, unique habitats, drinking water resources, and bathing water at the river basin level; (iii) the Conservation of Natural Habitats and Wild Fauna and Flora Directive, which maintains biodiversity while considering economic, social, cultural, and regional requirements; (iv) the Regulation on Prevention and Management of Invasive Alien Species, which lists and manages invasive alien species with significant adverse impacts; (v) the Regulation on Trade of Wild Fauna and Flora, which implements CITES regulations to protect species through regulated trade; and (vi) the Sustainable Blue Economy Strategy, which advances the European Green Deal’s objectives, supports renewable energy, decarbonizes maritime transport, and preserves biodiversity and landscapes by building green infrastructure.

**Table 1 marinedrugs-22-00427-t001:** International and European legislative frameworks relevant to the activities and processes of aquatic biorepositories.

Legislative Tool	Title	Objective	Parties
International Legislative Tools			
Nagoya Protocol https://www.cbd.int/abs/doc/protocol/nagoya-protocol-en.pdf	Access to Genetic Resources and the Fair and Equitable Sharing of Benefits Arising from their Utilization to the Convention on Biological Diversity [14]	Supplementary agreement to the Convention on Biological Diversity. Its objective is the fair and equitable sharing of benefits arising from the utilization of genetic resources.	Signatories can be found here. https://www.cbd.int/abs/nagoya-protocol/signatories
Cartagena Protocol https://bch.cbd.int/protocol/text	Biosafety to the Convention on Biological Diversity [15]	International agreement aiming to ensure the safe handling, transport, and use of LMOs	The list of parties can be found here https://bch.cbd.int/protocol/parties
CBD https://www.cbd.int/doc/legal/cbd-en.pdf	The Convention on Biological Diversity [16]	Three main objectives: (i) the conservation of biological diversity; (ii) the sustainable use of the components of biological diversity; and (iii) the fair and equitable sharing of the benefits arising out of the utilization of genetic resources.	The list of parties can be found here https://www.cbd.int/information/parties.shtml
UNCLOS https://www.un.org/depts/los/convention_agreements/texts/unclos/unclos_e.pdf	The United Nations Convention on the Law of the Sea [17]	Changes or introduces new concepts to the traditional law of the sea, such as (i) the maximum breadth of the territorial sea and the contiguous zone; (ii) the exclusive economic zone of coastal states in which they exercise sovereign rights and jurisdiction on all resource-related activities; (iii) a rule of reciprocal “due regard”; (iv) a series of articles dealing with the protection of the marine environment, setting out general principles and rules about competence for law-making, enforcement, and safeguards; and (v) provisions concerning marine scientific research.	Chronological lists of ratifications of and accessions and successions to the convention and the related agreements can be found here. https://www.un.org/Depts/los/reference_files/chronological_lists_of_ratifications.htm
CITES https://cites.org/sites/default/files/eng/disc/CITES-Convention-EN.pdf	Convention on International Trade in Endangered Species of Wild Fauna and Flora [18]	Since the trade in wild animals and plants crosses borders between countries, the effort to regulate it requires international cooperation to safeguard certain species from over-exploitation.	The list of parties can be found here. https://cites.org/eng/disc/parties/chronolo.php
CMS https://www.cms.int/sites/default/files/instrument/CMS-text.en_.PDF	Convention on the Conservation of Migratory Species of Wild Animals [19]	To conserve and take action to avoid any migratory species becoming endangered.	
Bern Convention https://rm.coe.int/1680078aff	Convention on the Conservation of European Wildlife and Natural Habitats [20]	The aims of this convention are to conserve wild flora and fauna and their natural habitats, especially those species and habitats whose conservation requires the cooperation of several states, and to promote the necessary cooperation. Emphasis is given to endangered and vulnerable species, including endangered and vulnerable migratory species.	The list of parties can be found here. https://www.coe.int/en/web/conventions/recent-changes-for-treaties?module=treaties-recent-changes&ddateDebut=05-05-1949&ddateStatus=10-03-2021&codeSignature=0&codeMatiere=8&numSTE=104
**Other European Union Legislative Tools**			
EU Marine Strategy Framework Directive (MSFD) https://eur-lex.europa.eu/legal-content/EN/TXT/PDF/?uri=CELEX:32008L0056&from=en	DIRECTIVE 2008/56/EC OF establishing a framework for in the field of marine environmental policy [21]	Aims (i) to protect the marine ecosystem and biodiversity; and (ii) to help EU countries achieve GES based on qualitative descriptors. The joint communication on international ocean governance proposes concrete measures at the international level, such as addressing environmental, fishery-related, and climate issues.	All EU Member States
EU Water Framework Directive (WFD) https://eur-lex.europa.eu/resource.html?uri=cellar:5c835afb-2ec6-4577-bdf8-756d3d694eeb.0004.02/DOC_1&format=PDF	Directive 2000/60/EC establishing a framework for in the field of water policy [22]	The protection of the aquatic ecology, the specific protection of unique and valuable habitats, the protection of drinking water resources, and the protection of bathing water at the river basin level. The directives for special habitats, drinking water areas and bathing water apply only to specific bodies of water (those supporting special wetlands; those identified for drinking water abstraction; and those generally used as bathing areas). In contrast, ecological protection should apply to all waters: the central requirement of the treaty is that the environment be protected to a high level in its entirety.	All EU Member States
Conservation of natural habitats and wild fauna and flora https://eur-lex.europa.eu/legal-content/EN/TXT/PDF/?uri=CELEX:31992L0043&from=EN	COUNCIL DIRECTIVE 92/43/EEC of 21 May 1992 on the conservation of natural habitats and of wild fauna and flora [23]	The main aim of this directive is to promote the maintenance of biodiversity, taking account of economic, social, cultural, and regional requirements. This directive contributes to the general objective of sustainable development, whereas the maintenance of such biodiversity may in certain cases require the maintenance, or indeed the encouragement, of human activities.	All EU Member States
Prevention and management of the introduction and spread of invasive alien species https://eur-lex.europa.eu/legal-content/EN/TXT/PDF/?uri=CELEX:32014R1143&from=EN	Regulation (EU) No. 1143/2014 of the EU Parliament and of the Council of 22 October 2014 on the prevention and management of the introduction and spread of invasive alien species [24]	The criteria for inclusion on the union list are the core instruments of the application of this regulation. To ensure the effective use of resources, those criteria should ensure that among the potential invasive alien species currently known, those that have the most significant adverse impact will be listed. The commission should submit to the committee established by this regulation a proposal for a union list based on those criteria within one year of this regulation entering into force. When proposing the Union list, the commission should inform that committee on how it considered those criteria. The criteria should include a risk assessment pursuant to the applicable provisions under the relevant agreements of the World Trade Organisation (WTO) on placing trade restrictions on species.	All EU Member States
Protection of species of wild fauna and flora by regulating trade therein https://eur-lex.europa.eu/legal-content/EN/TXT/PDF/?uri=CELEX:32019R0220&from=GA	COMMISSION REGULATION (EU) 2019/220 of 6 February 2019 amending Regulation (EC) No. 865/2006 laying down detailed rules concerning the implementation of Council Regulation (EC) No. 338/97 on the protection of species of wild fauna and flora by regulating trade therein [25]	The purpose of Commission Regulation (EC) No. 865/2006 (2) is to implement Regulation (EC) No. 338/97 and to ensure full compliance with the provisions of the Convention on International Trade in Endangered Species of Wild Fauna and Flora (CITES) (‘the convention’).	All EU Member States
Bern Convention https://rm.coe.int/1680078aff	Convention on the Conservation of European Wildlife and Natural Habitats [20]	This convention aims to conserve wild flora and fauna and their natural habitats, especially those species and habitats whose conservation requires the cooperation of several states, and to promote the necessary cooperation. Emphasis is given to endangered and vulnerable species, including endangered and vulnerable migratory species.	The list of EU and Non-EU Parties can be found here https://www.coe.int/en/web/conventions/recent-changes-for-treaties?module=treaties-recent-changes&ddateDebut=05-05-1949&ddateStatus=10-03-2021&codeSignature=0&codeMatiere=8&numSTE=104
Sustainable blue economy Strategy https://eur-lex.europa.eu/legal-content/EN/TXT/PDF/?uri=CELEX:52021DC0240&from=EN	On 17 May, the European Commission adopted the Communication on a new approach for a sustainable blue economy in the EU: “A Green Recovery for the Blue Economy–Transforming the EU’s Blue Economy for a Sustainable Future” [26]	The detailed agenda for the blue economy should help achieve the European Green Deal’s objectives and complement other recent commission initiatives on biodiversity, food, mobility, security, data, and more. For example, the blue economy contributes to climate change mitigation by developing offshore renewable energy to decarbonise maritime transport and greening ports. It will make the economy more circular by renewing standards for fishing gear design, ship recycling, decommissioned ship recycling, and decommissioning offshore platforms. The development of green infrastructure in coastal areas will help to preserve biodiversity and landscapes, while benefitting tourism and the coastal economy.	

The described legislative frameworks ensure that biorepository activities are conducted responsibly, sustainably, and in compliance with both international and European Union standards.

Adhering to Access and Benefit Sharing (ABS) frameworks, such as the Nagoya Protocol, can be challenging due to limited material availability and the insufficient dissemination of best practices for the ethical and equitable sharing of genetic resources. To assist biobanks and their users in complying with the ABS framework, the European Marine Biological Resource Centre (EMBRC) published the handbook “The EMBRC guide to ABS compliance: Recommendations for marine biological resource collections and user institutions” in 2020 [27]. The EMBRC handbook has gathered and synthesized a set of guidelines and recommendations, which are proposed as best practices for both collectors and users of genetic resources, to facilitate compliance with applicable legislation [27]. Implementing the handbook’s recommended best practices will achieve several objectives: (i) standardize operational procedures; (ii) assist users in adhering to existing legislative frameworks; (iii) mitigate liability issues for researchers; (iv) enhance transparency regarding the exploitation of genetic resources; (v) promote the fair and equitable sharing of the benefits derived from genetic resource utilization; (vi) support non-monetary benefit sharing for non-commercial research; (vii) reduce the unauthorized exploitation and misuse of genetic resources; and (viii) enhance legal certainty [27].

## 3. Recommendations, Guidelines, and Best Practices for the Sustainable Operation of an Aquatic Biomaterial Repository

Published guidelines for operating environmental specimen banks (ESBs), including aquatic biorepositories, are typically scattered across various documents and often treated as components of museum collections or research institutions’ facilities.

To differentiate ABR from BEB, ESB, NHC, and informal collections, the authors recommend the establishment of an overarching scientific committee comprising representatives from relevant fields associated with ABR. Due to the intersectoral nature of ABR’s procedures, it is important to maintain communication and collaboration among the various sections. Many workflow stages represent different fields of expertise, where each stage depends on the efforts of and results in another. Thus, it is crucial to have transparency in terms of actions, reporting, and intersectoral meetings to avoid missed opportunities or gaps in implemented actions. This can be achieved by forming a coordinating committee, tasked with researching current best practices, and developing a comprehensive manual for the successful and lawful establishment and management of ABR.

ABRs developed by research institutions, universities, or research NGOs can either appoint the most suitable scientists and experts as committee members or appoint them via a vote by the personnel. Each sector should be represented by at least one relevant expert, where the committee chair is appointed by a vote among the committee’s members. A well-organized committee representing all the involved sectors should have Articles of Association, outlining responsibilities, operations, liberties, obligations, and regulations. Funding for such a committee should be planned and allocated accordingly by the administrative authority of the organization.

Regular meetings can be held to coordinate each section to report on progress and to achieve the maximum exchange of information. The committee can use a sound monitoring and evaluation (M&E) plan (a document that helps to track and assess the results throughout the life of a project and establish the needed interventions) with set verifiable indicators. It must explain how the ABR’s achievements will be measured, providing accountability to the activities’ performance (i.e., indicators and means of verification). By consensus, the document will promote transparency and responsibility (who carries out what and when) while also providing standardization and coordination for applied methods/actions. A well-thought-out M&E plan is an invaluable tool that can guide the committee through the planning and implementation of set activities. Since an M&E plan is a living document, it may be revised and adapted after each evaluation stage based on the aims required to achieve the ABR’s goals. The evaluation of efforts translates to the systematic and objective assessment of an ongoing or completed action. It can determine the relevance and fulfilment of objectives, development efficiency, effectiveness, impact, and project sustainability. The review and evaluation of all activities can be performed annually by the committee and the M&E plan may be amended based on the assessed needs and gaps. Upon primary evaluation and reporting, the assigned committee can retrieve the financial resources and allocate the required funds where they are most needed.

A comprehensive compilation of procedures, methods, applications, requirements, and essential scientific knowledge was performed based on each stage of the ABR workflow to aid in establishing and managing aquatic biomaterial repositories. To streamline information, the workflow of ABR was divided into eight stages, as depicted in Figure 1 and Table 2:

Stage 1: sourcing—biospecimens are sourced ex situ (preserved or live specimens from Biological Resource Centres, Natural History Museums, etc.) or in situ (live specimens harvested/sampled from aquatic ecosystems). After collection, biospecimens are referred to as “vouchers.” At this stage, standardized biospecimen collection procedures are advised to ensure the replicability of the collection methods, if needed, and to safeguard the fair and equitable sharing of benefits arising from their utilization. Since sample collection needs careful planning, skilled personnel to carry it out, approval from the relevant authorities, and precautions regarding endangered species during sampling, it is to be viewed as one of the most crucial components of an ABR’s workflow. To meet the legal requirements, the depositors, providers, and recipients of genetic resources can consult Schneider, X. T. et al., 2022 [12].

Stage 2: identification—this is conducted before or after preservation and requires specialized scientific staff for taxonomical and molecular identification (e.g., genomic sequencing, PCR methods). It is recommended that standardised procedures be followed and applied to ensure correct species identification and prevent new and cryptic species from being misidentified. It is recommended that both methods be used for cross-checking. In a later stage, biomaterial will be collected from the identified species and their genomic information will be uploaded to libraries and datasets. Hence, it is imperative to have exhausted all routes that could lead to accurate taxonomic analysis.

Stage 3: biomaterial extraction and processing—these steps are conducted before or after preservation and require specialized scientific staff and laboratory equipment for tasks such as crude extract production, dereplication, biological screening (extracts, fractions, and pure compounds), the isolation and structural elucidation of aquatic natural products (NPs), the determination of absolute NP configuration, and biosynthetic pathway characterization through next-generation sequencing [28,29]. Best-practice protocols/standard operation procedures (SOPs) are suggested to maintain consistency between the datasets, ensure their replicability, and prevent the cross-contamination of samples. Chemically and biologically fractionated extracts are recommended for screening instead of more complex crude extracts which, coupled with modern high-throughput technology allowing the screening of thousands of samples, could skew screening results [3].

Stage 4: preservation—this is conducted before or after biospecimen identification and biomaterial processing. It involves the preservation of live aquatic biospecimens through cultivation (e.g., aquatic non-pathogenic microorganisms) and non-living biospecimens from ex situ and in situ sourcing. There is no universal preservation SOP. Thus, protocols and SOPs must be tailored to the specific group of biospecimens. Preservation techniques can differ among classes and clades of organisms, and it is recommended that the most suitable, replicable, efficient, and least destructive preservation method be used so as not to damage the genetic material or interfere with the success of the following workflow stages. Biodiversity collections must be flexible enough to adapt to the constantly shifting demands of science and technology.

Stage 5: the storage of biospecimens and biomaterial—this involves the long-term and short-term storage of aquatic biospecimens and biomaterial and requires specialized equipment based on the storage duration and applied SOPs. The use of quality management systems and accreditation to assess how best to apply ambient sample storage techniques is advised to ensure stability, reduce costs, improve handling logistics, and increase the efficiency of ABR procedures.

Stage 6: biospecimens and biomaterial databases—it is necessary to document all relevant information from stages 1 to 5. This required the creation of metadata, stored in digital libraries, inventories, or platforms, that are secure, easy to access, and regularly updated. It requires the implementation of data backup practices and metadata contingency plans to safeguard all collected data and information. The digitization of physical specimens provides visibility to an ABR’s biodiversity and biomaterial collection. A sound data management plan should be implemented that focuses on creating and maintaining a digital database in the form of a specialized marine natural product database. This will allow for the recording and tracking of all the data present within an ABR and ensure there are readily available comprehensive records.

Stage 7: the management of biospecimens and biomaterial—this involves the flow within biotechnology sectors and industries (pharmaceutical, nutraceutical, cosmetics, food, chemical, energy, environment, textile, naval, etc.). SOPs should govern the distribution and relocation of aquatic biospecimens (whether living or non-living, preserved or unpreserved) to other authorized facilities. At this stage, the ABR is transformed from a recipient of genetic resources into a provider of compounds derived from the collected biospecimens. Hence, it should pass its obligations on to the third party (private and public research centers, large companies, etc.) involved. The ABR must present the signed Material Transfer Agreements to the third party, and the latter must follow and respect the ABS and RRI frameworks [12].

Stage 8: dissemination and distribution—this involves sharing findings and applying new technologies and practices to the scientific and industrial communities. Scientific findings funded by public or European funds ought to be publicly accessible, and the free availability of publications and open science is recommended.

**Figure 1 marinedrugs-22-00427-f001:**
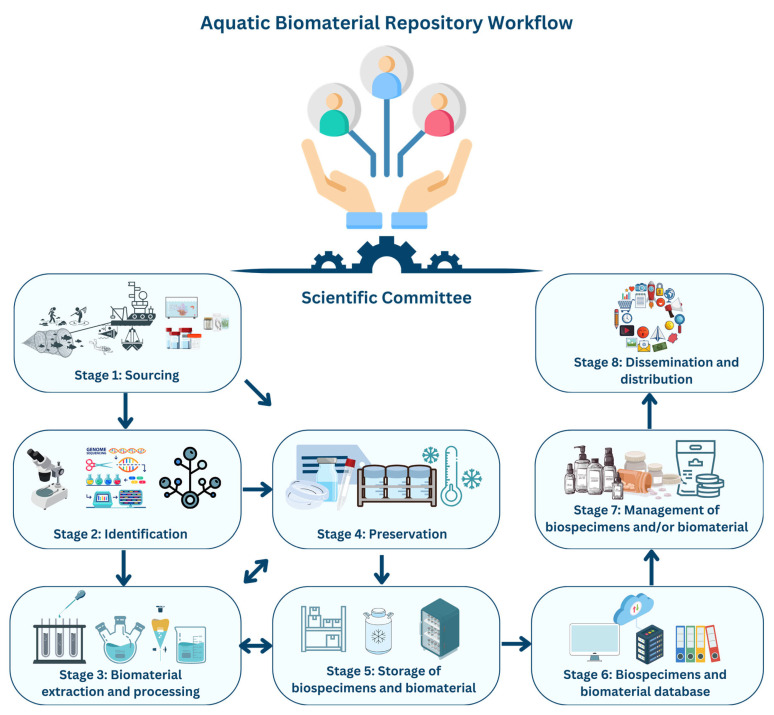
The workflow of an aquatic biomaterial repository (ABR). Stages 1 to 3 and 7 may be outsourced, with acquired biomaterial added to existing collections.

The guidelines and best practices for marine environmental biospecimen repositories cover various workflow stages, each supported by specific institutions and research centers. Table 2 compiles a list of reported guidelines and best practices applied in existing ABRs.

**Table 2 marinedrugs-22-00427-t002:** Guidelines and best practices are applied to establish and sustain the operation of ABRs.

Guidelines/Best Practices		
Workflow Stage	Aquatic Environment	Institution/Research Centre
Sourcing	Marine: in situ	1: The MarineBio Conservation Society (MarineBio) [30] 2: The Woods Hole Oceanographic Institution (WHOI) [31] 3–7: AQUACOSM [32,33,34,35,36] 8: European Marine Omics 9: Biodiversity Observation Network (EMO BON) [37] 9: The Scientific Committee on Oceanic Research (SCOR) [38] 10: The National Marine Biological Analytical Quality Control (NMBAQC) scheme [39]
	Freshwater: in situ	1: The Mediterranean Institute for Advanced Studies (IMEDEA) [40] 2: The United States Environmental Protection Agency (EPA) [41] 3: The United States Geological Survey (USGS) [42] 4: British Columbia Water Management Branch—MINISTRY OF ENVIRONMENT, LANDS AND PARKS. [43] 5–8: AQUACOSM [32,33,34,35]
Identification	See Section 5. Data Standardization—making genomic data discoverable	
Biomaterial Management	Marine–brackish–freshwater	1: Marine Biodiscovery, School of Chemistry and Ryan Institute, NUI Galway, Ireland [3] 2: European Marine Omics 9: Biodiversity Observation Network (EMO BON) [37]
Preservation	Preservation SOPs can be found in the referenced handbooks, guidelines and scientific journals/reviews/editorials in the Sourcing and Environmental Biospecimen Repositories sections.	
The storage of biospecimens and biomaterial	Storage SOPs can be found in the referenced handbooks, guidelines, and scientific journals/reviews/editorials in the sections for Sourcing and Environmental Biospecimen Repositories.	
Biospecimens and biomaterial database—data management	Marine–brackish–freshwater	1–3: Global Genome Biodiversity Network (GGBN) Data Standard specification [44,45,46] 4: The Genomic Standards Consortium (GenSC) [47] 5: EMBRC-ERIC Data Management Plan (2017) [48] 6: Association of European Marine Biological Laboratories Expanded. Deliverable D4.1 / ASSEMBLE Plus First Data Management Plan (2018) [49]
The management of biospecimens and biomaterial	Marine	1: The National Institute of Standards and Technology (NIST) [50] 2: ISBER [10]
The dissemination and exploitation of resources	Marine–brackish–freshwater	1: The National Institute of Standards and Technology (NIST) [50] 2: ISBER [10]
The dissemination of data and knowledge	Marine–brackish–freshwater	1: The European Biobanking and Biomolecular Resources Research Infrastructure (BBMRI) [51] 2: Ocean4Biotech platform [52]

Marine environmental biospecimen repositories include a diverse array of institutions dedicated to collecting, preserving, and studying marine specimens. Already established marine biorepositories and marine biomaterial repositories that apply the best practices are comprehensively described in the Appendix A. The National Institute of Standards and Technology (NIST) plays a key role with its Environmental Specimen Banking Programs and the NIST Biorepository [50], formerly known as the Marine Environmental Specimen Bank. The ISBER provides best practices and guidelines for these repositories [10]. In Uganda, the BUILD: Boosting Uganda’s Investment in Livestock Development initiative focuses on biorepository workflows to boost investment in livestock development [53]. The EMBRC and its extended network, EMBRC-ERIC, support European marine biological research [54]. The National Marine Mammal Tissue Bank (NMMTB), which is under the National Oceanic and Atmospheric Administration (NOAA) Fisheries, is dedicated to the preservation of marine mammal tissues for research and conservation purposes [55]. The Smithsonian National Museum of Natural History’s Global Genome Initiative (GGI) aims to collect and preserve genomic materials from across the world’s biodiversity [56]. Ireland’s National Marine Biodiscovery Laboratory, under the Marine Institute European Maritime and Fisheries Fund (EMFF) 2014–2020, focuses on biomaterials research [57], while the Marine Institute Foras na Mara serves as Ireland’s national agency for marine research, technology, development, and innovation [58]. These repositories collectively contribute to the global effort to understand and preserve marine biodiversity, supporting a wide range of scientific research and conservation initiatives.

An important reference is the “BEST PRACTICES: Recommendations for Repositories, Fourth Edition” (2018) by ISBER [10]. This guidance document consolidates the collective expertise of ISBER members and aims to disseminate effective strategies, policies, and procedures necessary for the successful operation of biorepositories [10]. While adherence to ISBER Best Practices is voluntary, these guidelines assist biorepositories in meeting regulatory and accreditation standards. Maintaining biospecimen integrity and quality over extended periods poses a significant challenge for biorepositories. Implementing a robust quality management system encompassing quality assurance and control is essential. This involves validating and qualifying instruments, reagents, and methods to meet established standards. Ensuring biospecimen and data security is equally critical. Best practices for data security include regular backups on remote secure servers and the strict control of user access based on roles. Secure, cloud-based biobanking Laboratory Information Management System (LIMS) platforms ensure comprehensive data security with user authentication, tiered access controls, built-in firewalls, and encrypted data storage and transmission [10].

## 4. Best Practices for the Dissemination of Data and Knowledge—Collaborative Networks

It is well known that useful information and knowledge gained from biological studies, especially aquatic studies, are frequently not extensively shared with stakeholders such as industrial actors, researchers, the general public, policy makers, and environmental experts. The efficient and rational exploitation of ocean resources requires direct interaction among stakeholders. This practice has been limited, with only a few programs facilitating transdisciplinary interaction [59].

One of the programs facilitating stakeholder interaction and knowledge dissemination was COST Action CA18238—European Transdisciplinary Networking Platform for Marine Biotechnology, known as Ocean4Biotech. Described as an international, unique, and inclusive network, Ocean4Biotech brought together experts from diverse fields, including exact and natural sciences, social sciences, and humanities. Participants in this initiative collaborated to foster marine biotechnology and bioeconomy sustainably, creating a spill-over effect through shared experiences [59]. Ocean4Biotech seeks to engage experts from various disciplines contributing to biodiscovery and biotechnology under the RRI, spanning fields such as food science, agriculture, pharmacology, medicine, environmental protection, data science, omics, law, and policymaking. Moreover, the network served as a platform for transferring knowledge from traditional academic institutions and research centers to industrial stakeholders, policymakers, and the broader public [59]. To enhance interaction within the marine biotechnology community, Ocean4Biotech developed an online platform. This platform allowed researchers and community members to showcase their expertise, fostering new collaborations and serving as a search tool for potential partnerships [60].

The Global Genome Biodiversity Network (GGBN) offers its members a platform for biodiversity biobanks, enabling them to make their DNA and tissue collections accessible for research [61]. By advocating for the adoption of best standards and practices in genetic collections, all members adhere to harmonized methods and practices. This ensures consistent quality standards for DNA and tissue collections and the preservation, utilization, and exchange of materials in compliance with national and international laws and conventions [10].

The European Marine Omics Biodiversity Observation Network (EMO BON) was established as part of the EMBRC initiative to bolster individual observatories and integrate them into a centrally coordinated network. Its primary goal is to establish new observatory stations along European coastlines, focusing on generating comprehensive data on the composition of biodiversity [62].

## 5. Data Standardization—Making Genomic Data Discoverable

Global reference lists of genomic information are crucial for understanding biodiversity and ecosystems, which is achievable only via the integration of morphological and molecular methods. The establishment of the GGBN Data Portal provided a platform that links biodiversity repositories, sequence databases, and research findings, integrating genomic sample-vouchered specimens, sequence data, and publications [45]. This enhances the discoverability and utilization of genomic samples and data.

In molecular-based identification, comprehensive reference databases with detailed documentation are essential for automated sequence comparisons, such as BLAST (Basic Local Alignment Search Tool) [63], which compares sequences against primary sequence databases like the Nucleotide Collection (nt) operated by the International Nucleotide Sequence Database Collaboration (INSDC) [64].

The GGBN Data Portal [44,45] employs a set of terms and controlled vocabularies specifically designed to represent sample facts, focusing on molecular terms from Minimum Information about any (X) Sequence (MIxS), Minimum Information about a MARKer gene Sequence (MIMARKS), and Minimum Information about a Genome Sequence (MIGS) [46]. It complements standards like Access to Biological Collection Data (ABCD) and Darwin Core (DwC). It incorporates SPREC (Standard PRE analytical Codes) and elements of BRISQ (Biospecimen Reporting for Improved Study Quality) [45]. Another significant public domain for the sharing of nucleotide sequences and associated metadata is the International Nucleotide Sequence Database Collaboration (INSDC), comprising databases such as the DNA Data Bank of Japan, the European Nucleotide Archive, and GenBank [64].

Additionally, the Genomic Standards Consortium (GSC), established in 2005, promotes the discoverability of genomic data through international community-driven standards. It supports various initiatives such as FAIRsharing, aimed at promoting data and metadata standards related to databases and data policies; Micro B3 (Biodiversity, Bioinformatics and Biotechnology), focusing on marine microbial bioinformatics platform development; MIxS-BE (MIxS for indoor metagenomics), a package for describing microbial communities in built environments; M5 (Metagenomics, Metadata, MetaAnalysis, Models and MetaInfrastructure); and the MIxS GSC Project [46,47], which sets core standards for describing genomes, metagenomes, and gene marker sequences.

## 6. The Interconnection of Biodiscovery and Aquatic Biodiversity Conservation through Biorepositories

According to Reddy et al.’s (2021) publication “Marine Biodiscovery in a Changing World” [3], there are three recognized opportunities for biorepositories to help mitigate the loss of aquatic biota: (i) The establishment of regional marine biomaterial repositories can address some of the recognized threats, as areas identified as valuable for biodiscovery and biotechnological use tend to receive greater protection from external dangers. (ii) The development of new technologies for the screening and dereplication of the biomaterials will also contribute to addressing some of the issues. (iii) Biorepositories can propose significant changes to make the concept of marine biodiscovery more inclusive and central to the development of the three pillars of sustainability: the blue economy, environmental challenges, and social impacts.

As mentioned above, the primary goal of a biorepository is to collect, catalog, store, preserve, and manage biospecimens and biomaterials. Through these activities, a greater understanding of local biodiversity is achieved, and a permanent database is created to archive indigenous, endemic, and invasive species along with their genomic information. Cataloging local biodiversity is an integral part of preserving biodiscovery, especially since many aquatic species in various countries remain unknown and undiscovered [65]. Discovering previously unknown aquatic organisms and recording known species within a single unit benefits human sustainability. Aquatic organisms are a nutrient-rich and stable food source for humans, significantly contribute to natural product chemistry, and are critical for drug discovery.

Exploring new ecosystems or focusing biodiscovery research on non-model organisms may lead to the discovery of new natural products that promote human health [3]. Collecting organisms from a wide range of species families allows researchers and scientists to assess local biodiversity and conduct surveillance and monitoring schemes that facilitate conservation efforts. Environmental protection and restoration often result from policies and political commitments from higher governmental levels. Newly discovered aquatic bioresources of value to the biotechnological industry could provide significant leverage and incentives for national and subnational governments to draft conservation policies and strategies for the ecosystems hosting them. Finally, databases and metadata gathered and processed by biomaterial repositories based on open science policies will greatly benefit the exchange of knowledge regarding species distribution, classification, evolution, and adaptation [3].

## 7. The Role of Biomaterials in Promoting Conservation

Biotechnology and the genomic and transcriptomic data gathered from biomaterials significantly contribute to conservation efforts. These data can be used to assess the potential of populations to adapt to new challenges such as climate change, invasive species, and shifts in distribution, among others [66]. Additionally, the in vitro methods used to preserve and conserve the genetic material of rare and threatened species in aquatic biomaterial repositories promote gene diversity conservation and facilitate their reintroduction into the wild, supported by novel techniques and methods such as de-extinction [66,67]. Molecular methods performed in biomaterial repositories, including metabarcoding and next-generation sequencing (NGS), provide insight into the (meta)genomes, (meta)transcriptomes (cDNA), and (meta)barcodes of individuals, populations, and communities [64]. These techniques open new avenues for studying and describing the biodiversity and taxonomy of organisms, microorganisms, and viruses, offering insights into their ecology from both the past and present [68]. In summary, conservation genetics has yielded important information on the dynamics of endangered populations, allowing the application of ‘conservation prior’ in managing aquatic populations [68].

## 8. Benefits of Establishing a Biomaterial Repository for Local Authorities and Communities

Biomaterial repositories offer significant advantages for local authorities, requiring minimal running capital and initial investment. The initial costs of establishing an aquatic biomaterial repository include several components: (i) facilities to house biospecimens (alive or preserved), collected tissues, and molecular materials; (ii) specialized equipment for collecting, storing, processing, extracting, classifying, documenting, and managing the collection; (iii) personnel dedicated to its activities; and (iv) research and development procedures [3]. However, despite these upfront investments, the potential for direct and indirect future profits can justify the expenditure. This investment can also facilitate a trade-off between costly and time-consuming research expeditions and the maintenance of redundant collections [3]. Additionally, the benefits span biological, chemical, and conservational aspects and contribute to public health, the economy, community development, and employment [3]. Specifically, the benefits include the following: (i) biodiversity monitoring—the collected biospecimens represent local species, allowing scientists to evaluate changes in the composition and distribution of indigenous and/or invasive species; (ii) research capacity building—developing countries can greatly benefit from the operation of a biomaterial repository, enhancing their research capacities in fields such as taxonomy [69]; (iii) public health—marine-derived compounds can efficiently address emerging or re-emerging diseases, with biomaterials readily available for research and development; (iv) economic development—local communities benefit from new opportunities in the blue economy, intellectual property from active biomaterials, and stored molecular entities; (v) industry benefits—relevant industries and companies may benefit from the royalties and licenses of newly developed biomolecules; (vi) employment opportunities—establishing a biomaterial repository can create new job opportunities, requiring specific qualifications and offering a valuable field of employment; and (vii) social and educational benefits—technology and knowledge transfer to local scientists can provide significant social benefits, fostering international cooperation and enriching communities with diverse research and scientific expertise [3].

## 9. Key Points for Sustainable Implementation and Maintenance of ABRs

Creating and managing an ABR requires expertise, specialized personnel, and advanced technological and biochemical methods. ABRs collect, process, store, and distribute biomaterials, such as DNA, RNA, proteins, enzymes, biopolymers, and natural products from various sources, including natural history collections, bioprospecting results, and research activities. The establishment and sustainable operation of ABRs are critical for addressing the ongoing loss of aquatic biodiversity and supporting scientific, economic, and environmental goals. By implementing standardized practices, complying with regulations, embracing technological advancements, fostering collaboration, and engaging local communities, ABRs can significantly contribute to conservation efforts and biotechnological innovation. These repositories protect invaluable genetic resources and promote sustainable development and public health, highlighting their multifaceted importance in today’s world. For sustainable development and the maintenance of ABRs, key points for its successful implementation are suggested (Figure 2): (i) The standardization of practices—developing and implementing standardized protocols for the collection, storage, processing, and distribution of biospecimens can enhance the reliability and reproducibility of research, as recommended in stages 1, 2, 4 and 5. Furthermore, adopting ISBER’s Best Practices and creating comprehensive guidelines specific to ABRs can address inconsistencies. (ii) Regulatory compliance and ethical practices—ensuring compliance with international agreements, such as the Nagoya Protocol, ABS, and national regulations outlined in stages 1 and 7, is crucial. This involves understanding and adhering to legislation on genetic resource sharing, which handbooks like the EMBRC guide can also facilitate. (iii) Technological advancements updates—investing in new technologies for genomic sequencing, data management, and biomaterial preservation, as suggested in stages 2, 3, 4 and 5, can improve the efficiency and effectiveness of ABRs. These advancements can help in the rapid identification and processing of biospecimens. (iv) Collaborative networks—establishing and participating in networks can foster collaboration, knowledge exchange, and resource sharing among researchers, policymakers, and industry stakeholders. This enhances the collective ability to address biodiversity loss and environmental challenges. (v) Data standardization and accessibility—creating comprehensive databases and metadata platforms for genomic and ecological data (stage 6—digitalization of biospecimens and compounds) can improve data discoverability and utility. Ensuring that data are accessible to researchers globally via open access and open science supports collaborative research and conservation efforts. (vi) Local and community engagement—engaging local communities in conservation and biorepository activities can enhance public awareness and support biodiversity preservation. Community involvement in monitoring and managing local ecosystems can provide valuable data and strengthen conservation initiatives. (vii) Quality management systems—implementing robust quality management systems that include quality assurance and control measures ensures the integrity and security of biospecimens and data. Regular validation and qualification of instruments and methods are essential components of these systems. (viii) Funding and investment—securing funding for the establishment of sustainable ABRs is crucial. This can be achieved through public and private investments, grants, and partnerships with industry stakeholders who benefit from the biotechnological applications of aquatic biomaterials.

Obtaining permits to collect biological samples in compliance with the Nagoya Protocol is an essential aspect of ensuring the fair and equitable use of genetic resources. However, the bureaucracy involved in these requests often becomes a significant obstacle to scientific research [12,13]. In many cases, the national focal points (NFPs) responsible for evaluating and issuing these permits fail to respond in a timely manner, undermining the effectiveness of research and the adherence to international standards. To mitigate these challenges and promote the more effective functioning of the NFPs, a simplified digital platform is proposed to centralize and expedite the application process and permit sample collection. This platform would operate globally, with a link connecting users with the ABS website, allowing researchers to submit their requests in a standardized format, ensuring that all necessary information is appropriately documented and readily available for analysis.

The implementation of a digital platform would bring several benefits. Firstly, it would significantly reduce response times by automating bureaucratic steps and facilitating the flow of information between researchers and the NFPs. Secondly, it would provide greater transparency in the process, allowing applicants to track the status of their requests in real time and reducing the uncertainty associated with waiting for responses. Additionally, it would create a consolidated and accessible database, which would help to identify common bottlenecks in the process and promote continuous improvements. Simplifying this process would encourage compliance with the Nagoya Protocol and facilitate international collaboration and the sharing of biological resources, promoting more open and inclusive science. By improving the efficiency of NFPs and ensuring timely responses, this platform could drive the advancement of research and contribute to a more sustainable and equitable use of natural resources.

The development of a simplified digital platform for sample collection permit requests represents a practical and effective solution to current bureaucratic challenges, strengthening compliance with the Nagoya Protocol and ensuring the continuous progress of science in a global context of sustainability and cooperation.

Additionally, the creation of a global and free digital ABR platform for the registration of biospecimens (biological samples) and biomaterials, inspired by models like the NCBI and repositories such as GenBank, represents a strategically essential step in advancing global biotechnology and drug discovery research. This platform would allow the registration, cataloging, and availability of information on biological collections globally, promoting greater transparency and accessibility in sharing scientific data. By ensuring compliance with internationally established ethical and quality standards, the database would help to mitigate risks associated with inadequate practices and safeguard the integrity of research conducted using these resources. To further enhance the effective management and development of this digital platform, it is proposed that the platform layout and guidelines be established by a scientific committee comprising representatives from relevant fields associated with ABRs. This committee would be tasked with researching current best practices and developing a comprehensive manual for the successful and lawful establishment and management of ABRs. Such an initiative would contribute to harmonizing practices across different countries and institutions, fostering a more consistent and standardized approach to collecting, storing, conserving, handling, and distributing biological materials. 

This digital ABR platform would create a cohesive framework that promotes transparency, ensures compliance with international standards, and drives adherence to best practices. Free access to a global catalog of collections would facilitate collaboration among institutions from different countries, enhance efficiency, and reduce the duplication of efforts in the collection and analysis of samples. Scientists and developers of biotechnological products could consult this database to quickly identify available biospecimens and biomaterials and their characteristics, enabling them to select the most suitable samples for specific studies.

This global digital repository would encourage the development of new research techniques, enabling large-scale comparative analyses and fostering scientific innovation. In the long term, the platform could serve as a central hub for knowledge exchange and advance biological and medical sciences, benefiting public health and global well-being. Digital ABRs align with open and collaborative science principles, promoting free and equitable access to scientific knowledge. They would also strengthen trust in the quality and reproducibility of globally available scientific data, thereby fostering greater collaboration and advancing scientific research in aquatic biomaterial repositories.

Together, these initiatives would create a robust infrastructure that complies with international regulations and sets a new standard for the ethical and efficient management of biospecimens worldwide.

## 10. Conclusions

Despite recognizing the critical importance of aquatic biospecimen and biomaterial repositories, the thorough literature investigation conducted within the COST ACTION Ocean4Biotech CA18238 framework has revealed a significant gap. Specifically, there is a lack of standardized procedures, guidelines, and best practices applicable to the various workflow stages of ABRs and biomaterial repositories. Across Europe, notable heterogeneity exists in the activities and operational procedures of biorepositories. This variability may impede the ethical, equitable, and beneficial sharing of genetic resources and pose challenges in terms of implementing and auditing standardized methods. Efforts to establish ABRs must navigate diverse regulatory landscapes and adhere to international agreements such as the Nagoya Protocol and various EU directives. Compliance with these frameworks ensures equitable benefit sharing and the sustainable utilization of genetic resources, essential for fostering international cooperation and environmental stewardship.

The establishment and operation of ABRs represent a crucial step towards biodiversity conservation and sustainable biotechnological advancement. The urgent need to protect aquatic ecosystems and the potential loss of invaluable species underscores the importance of ABRs in preserving biological diversity and facilitating scientific research. ABRs serve as repositories for a wide array of biomaterials derived from aquatic organisms, ranging from DNA and RNA to biopolymers and natural products. These repositories safeguard biological resources and support research in fields such as drug discovery, natural product chemistry, and environmental biotechnology. By compiling and adhering to standardized best practices, ABRs can ensure the integrity, accessibility, and ethical use of collected specimens and data. 

Compiling this comprehensive document on ABR key points fills gaps in standardized procedures and promotes the establishment of new repositories globally. By providing a roadmap for the collection, processing, storage, management, and distribution of aquatic biomaterials, this document empowers local authorities and scientific communities to initiate and sustain ABRs effectively. Furthermore, the socio-economic benefits of ABRs extend beyond scientific research, offering opportunities for economic development, public health improvements, and community capacity building. These repositories contribute to local economies through biotechnological innovations and engage communities in conservation efforts and educational initiatives. By embracing best practices and fostering international collaboration, ABRs can play a transformative role in the future of marine biotechnology and environmental conservation.

The proposed initiatives, a simplified digital platform for obtaining permits in compliance with the Nagoya Protocol and a global, free digital ABR platform for registering biospecimens and biomaterials, represent a transformative approach to overcoming current bureaucratic barriers and advancing scientific research. By streamlining the permit application process and creating a centralized platform that connects directly to the ABS website, researchers will be able to submit requests more efficiently and transparently, reducing delays and enhancing compliance with international standards. The global digital ABR platform for registering biological materials will further drive this transformation by enabling the worldwide cataloguing and ensuring the availability of information from biological collections. This will promote transparency, accessibility, and adherence to best practices in terms of collecting, storing, and distributing biological materials while fostering international collaboration and innovation. Establishing a scientific committee to guide the development and management of the digital repository will ensure that practices are harmonized across countries and institutions, leading to a more standardized and consistent approach to the management of biospecimens.

Combined, these initiatives will create a cohesive framework that not only aligns with the principles of open and collaborative science but also enhances the quality and reliability of, as well as global trust in, scientific data. Ultimately, these efforts will provide significant benefits to public health, environmental sustainability, and global scientific progress, setting a new standard for the ethical and efficient management of genetic resources and biospecimens worldwide.

## Figures and Tables

**Figure 2 marinedrugs-22-00427-f002:**
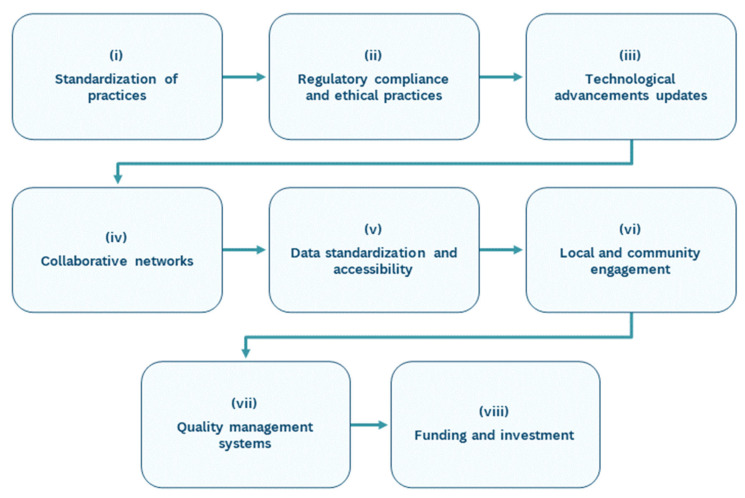
Key points for the successful establishment of an ABR.

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
