# Peer review of "Aquatic Biomaterial Repositories: Comprehensive Guidelines, Recommendations, and Best Practices for Their Development, Establishment, and Sustainable Operation"

_marinedrugs, 2024, doi:10.3390/md22090427_

Round 1

Reviewer 1 Report

Comments and Suggestions for Authors

The manuscript of Tourapi et al. is a comprehensive review of existing guidelines for the collection, storage and management of marine bioresources through Aquatic Biomaterial Repositories (ABRs). This is an extremely important topic given global efforts to maintain and protect biodiversity and also recognize the rights of indigenous cultures to their biological heritage. 

Overall I found the manuscript to be well written and expansive in scope. It lays out current methodologies for the establishment and operation of ABRs, it summarizes the many different guidelines published in different jurisdictions for control of such repositories, and it closes with some recommendations.

My main concern with the manuscript is that given the wide scope and the large amount of literature covered, and that many of the guidelines are published by specific stakeholder groups under many project names, there are a large number of abbreviations used throughout and I found it difficult to keep all the different organizations, guidelines and projects in mind.

I was also a bit disappointed in the recommendations of the manuscript. Given the amount of literature covered, section 9 with key recommendations is rather short and is in my mind, somewhat vague; the recommendations are largely aspirational without much substance. I wonder if the authors can add more to this section to provide the reader with concrete recommendations of best practice, with reference to specific examples for each of their eight key points.

The other item that I feel needs more attention is the final paragraph of the conclusion (section 10). The authors recommend the establishment of a committee to oversee ABRs, which is an interesting notion. There are some recommendations about the scope of such a committee but again, a lack of specific detail. How would such a committee be organized? Who would fund it? What powers and jurisdiction would it have? How would membership be arranged and controlled? This is a very interesting suggestion and one that could be extremely valuable, but I feel the suggestion is a surprise at the end of the conclusion and leaves the reader with more questions than answers.

In way of corrections, I have uploaded a marked up PDF with minor editorial comments. One final editorial request, tables 2 and 3 include references to the primary literature that are also included in the reference list, but the citations receive their own reference number in the table which differs from that in the final bibliography. For example, Santi et al. is cited three times in the manuscript, listed as ref 2 and 8 in the tables and 37 in the reference list. Please use the primary reference list citation numbering for the tables as well. 

Comments on the Quality of English Language

English is largely fine, minor corrections needed.

Author Response

We warmly thank you for the time dedicated for the review of our manuscript. We consider that all the reviewer suggestions have supported to improve its readability and pinpoint the main message and the most critical factors for developing and maintaining biorepositories. We provide a point-by-point reply to each suggestion and upload a revised manuscript with track changes.

Reviewer 1

The manuscript of Tourapi et al. is a comprehensive review of existing guidelines for the collection, storage and management of marine bioresources through Aquatic Biomaterial Repositories (ABRs). This is an extremely important topic given global efforts to maintain and protect biodiversity and also recognize the rights of indigenous cultures to their biological heritage. 

Overall I found the manuscript to be well written and expansive in scope. It lays out current methodologies for the establishment and operation of ABRs, it summarizes the many different guidelines published in different jurisdictions for control of such repositories, and it closes with some recommendations.

My main concern with the manuscript is that given the wide scope and the large amount of literature covered, and that many of the guidelines are published by specific stakeholder groups under many project names, there are a large number of abbreviations used throughout and I found it difficult to keep all the different organizations, guidelines and projects in mind.

  • A table of abbreviations was created and is submitted in the supplementary material

I was also a bit disappointed in the recommendations of the manuscript. Given the amount of literature covered, section 9 with key recommendations is rather short and is in my mind, somewhat vague; the recommendations are largely aspirational without much substance. I wonder if the authors can add more to this section to provide the reader with concrete recommendations of best practice, with reference to specific examples for each of their eight key points.

  • Section 9 has been extensively amended, and specific recommendations based on each point (2-3 sentences) have been added.

The other item that I feel needs more attention is the final paragraph of the conclusion (section 10). The authors recommend the establishment of a committee to oversee ABRs, which is an interesting notion. There are some recommendations about the scope of such a committee but again, a lack of specific detail. How would such a committee be organized? Who would fund it? What powers and jurisdiction would it have? How would membership be arranged and controlled? This is a very interesting suggestion and one that could be extremely valuable, but I feel the suggestion is a surprise at the end of the conclusion and leaves the reader with more questions than answers.

  • Thank you for this comment. We consider that the committee creation suits better to the Section 9 Recommendations Section and extra information about the role of the committee has been added. We did not provide possible funding mechanisms for this scientific committee as it relates to the level of its jurisdiction. For instance, at the global level, it could be covered by other initiatives such as NCBI, and at the continental European level from entities such as the European Molecular Biology Laboratory and the European Commission.

In way of corrections, I have uploaded a marked up PDF with minor editorial comments. One final editorial request, tables 2 and 3 include references to the primary literature that are also included in the reference list, but the citations receive their own reference number in the table which differs from that in the final bibliography. For example, Santi et al. is cited three times in the manuscript, listed as ref 2 and 8 in the tables and 37 in the reference list. Please use the primary reference list citation numbering for the tables as well. 

  • The editorial comments have been addressed and the citations reorganized using a literature management software

Reviewer 2 Report

Comments and Suggestions for Authors

Aquatic Biomaterial Repositories: Comprehensive guidelines, recommendations and best practices for their development, establishment and sustainable operation

The review emphasizes the urgent need to establish and maintain aquatic biomaterial repositories (ABRs) to protect vital aquatic organisms. It underscores the importance of standardized procedures and adherence to regulations like the Nagoya Protocol for responsible resource management. Properly managed ABRs can significantly contribute to marine biotechnology and environmental conservation, mitigating the risks of species extinction. After reviewing the manuscript, I would like to leave some comments in order to improve their quality.

Line 32: Please, “a stounding” should be “astounding”.

Line 35: Please, correct “threads” should be “threats”.

Lines 39-40: Clarify the list with proper punctuation and conjunctions. Add “and” before the last item in the list for clarity. For example, “(x) geological events, and (xi) climate change and severe weather conditions”.

Line 35: “biota loss” is a bit vague. Consider “loss of biodiversity”.

Line 42: Please, consider “biotechnology fields and are crucial contributors to environmental and human health.” This sentence is awkwardly phrased.

Line 47: Redundancy in the phrase “such as, as a source of food”.

Line 45: The sentence “unique structures and bioactivities provide new avenues for the bioeconomy” can be made more concise. I mean, for example, “unique structures and bioactivities open new avenues for the bioeconomy”.

Line 48: Please, the phrase “Therefore, the need to protect aquatic ecosystems and establish viable biorepositories for biomaterials and bioactive compounds has never been more critical.” is too wordy. I would suggest that authors changing it for something like, “Thus, protecting aquatic ecosystems and establishing biorepositories for biomaterials and bioactive compounds is crucial.”

Line 51: “multi-layer process” should be “multi-layered process”.

Line 53: Remove “as a repository that”, is redundant.

Line 63: I would introduce the sentence saying something like, “according to…”.

Line 86: Please, correct “for biorepository development”.

Line 113: Use abbreviation consistently after the first mention.

Table 1 and Table 2 could be summarized, it is too long.

Line 175: Change to “ABR workflow” (singular, as it's referring to one workflow).

Line 184: Please, change to “DNA extraction” (since “molecular” is redundant here).

Lines 263-265: The phrase “knowledge acquired, and informative data gathered” can be improved.

Lines 267-269: I would say, “Efficient and rational exploitation of ocean resources requires direct interaction among stakeholders. This practice has been limited, with only a few programs facilitating such transdisciplinary interaction [62]”.

Lines 293-295: The phrase “fosters collaboration among members” is slightly redundant with “ensuring consistent quality standards”.

The manuscript lacks figures, authors could consider adding figures to illustrate the information. Tables should be more elaborated and concise. I do not find the usefulness of Table 3, it has a complex readability and does not fit in the manuscript.

The conclusion section could be expanded to include future perspectives, offering a broader insight into potential developments and directions.

The manuscript provides valuable insights into the environmental sustainability of aquatic biospecimen repositories, identifying significant gaps in standardized procedures and guidelines. The variability in practices across Europe poses challenges for ethical and effective resource sharing. It is also necessary to check the references carefully and to make them consistent. Despite the need for MINOR revisions to improve clarity and coherence, the manuscript’s contributions are commendable. I encourage the authors to address the suggested changes to enhance the manuscript's impact.

Comments on the Quality of English Language

Above

Author Response

Reviewer 2

Aquatic Biomaterial Repositories: Comprehensive guidelines, recommendations and best practices for their development, establishment and sustainable operation

The review emphasizes the urgent need to establish and maintain aquatic biomaterial repositories (ABRs) to protect vital aquatic organisms. It underscores the importance of standardized procedures and adherence to regulations like the Nagoya Protocol for responsible resource management. Properly managed ABRs can significantly contribute to marine biotechnology and environmental conservation, mitigating the risks of species extinction. After reviewing the manuscript, I would like to leave some comments in order to improve their quality.

Line 32: Please, “a stounding” should be “astounding”.

Line 35: Please, correct “threads” should be “threats”.

Lines 39-40: Clarify the list with proper punctuation and conjunctions. Add “and” before the last item in the list for clarity. For example, “(x) geological events, and (xi) climate change and severe weather conditions”.

Line 35: “biota loss” is a bit vague. Consider “loss of biodiversity”.

Line 42: Please, consider “biotechnology fields and are crucial contributors to environmental and human health.” This sentence is awkwardly phrased.

Line 47: Redundancy in the phrase “such as, as a source of food”.

Line 45: The sentence “unique structures and bioactivities provide new avenues for the bioeconomy” can be made more concise. I mean, for example, “unique structures and bioactivities open new avenues for the bioeconomy”.

Line 48: Please, the phrase “Therefore, the need to protect aquatic ecosystems and establish viable biorepositories for biomaterials and bioactive compounds has never been more critical.” is too wordy. I would suggest that authors changing it for something like, “Thus, protecting aquatic ecosystems and establishing biorepositories for biomaterials and bioactive compounds is crucial.”

Line 51: “multi-layer process” should be “multi-layered process”.

Line 53: Remove “as a repository that”, is redundant.

Line 63: I would introduce the sentence saying something like, “according to…”.

Line 86: Please, correct “for biorepository development”.

Line 113: Use abbreviation consistently after the first mention.

Table 1 and Table 2 could be summarized, it is too long.

Line 175: Change to “ABR workflow” (singular, as it's referring to one workflow).

Line 184: Please, change to “DNA extraction” (since “molecular” is redundant here).

Lines 263-265: The phrase “knowledge acquired, and informative data gathered” can be improved.

Lines 267-269: I would say, “Efficient and rational exploitation of ocean resources requires direct interaction among stakeholders. This practice has been limited, with only a few programs facilitating such transdisciplinary interaction [62]”.

Lines 293-295: The phrase “fosters collaboration among members” is slightly redundant with “ensuring consistent quality standards”.

  • All these changes have been addressed in the revised manuscript.

The manuscript lacks figures, authors could consider adding figures to illustrate the information. Tables should be more elaborated and concise. I do not find the usefulness of Table 3, it has a complex readability and does not fit in the manuscript.

  • Thank you for your suggestion. A Graphical abstract has been produced along with a graph about the importance of biorepositories for marine resources and the Section 9 workflow. Table 3 was moved to the Supplementary material

The conclusion section could be expanded to include future perspectives, offering a broader insight into potential developments and directions.

  • The Conclusion Section has been reviewed and partly expanded.

The manuscript provides valuable insights into the environmental sustainability of aquatic biospecimen repositories, identifying significant gaps in standardized procedures and guidelines. The variability in practices across Europe poses challenges for ethical and effective resource sharing. It is also necessary to check the references carefully and to make them consistent. Despite the need for MINOR revisions to improve clarity and coherence, the manuscript’s contributions are commendable. I encourage the authors to address the suggested changes to enhance the manuscript's impact.

  • Thank you for your comment. The list of references has been updated.